# New Approach in COD Fractionation Methods

**Ewelina Płuciennik-Koropczuk \*** and **Sylwia Myszograj**

Faculty of Civil Engineering, Architecture and Environmental Engineering, Institute of Environmental Engineering, University of Zielona Góra, 15 Prof. Z. Szafrana St, 65-516 Zielona Góra, Poland
* Correspondence: E.Plucienik@iis.uz.zgora.pl; Tel.: +48-68-3282574

**Abstract:** Conventional quality parameters such as Chemical Oxygen Demand (COD) or Biochemical Oxygen Demand (BOD) give information about the quantity of organic matter present in wastewater, but do not give a clear indication of the biodegradability of the pollutants flowing in the WWTP. Detailed knowledge can be obtained by dividing the total COD into fractions. Fractionation and balancing of COD can be determined in various ways and with varying accuracy. Good wastewater characteristics are obtained on the basis of COD fractionation in accordance with ATV-A 131 guidelines, especially when the wastewater characteristics are in high compliance with the assumptions of the method. The article proposes a modification of the ATV-A131 method that increases the accuracy of determining the COD fraction. In order to reduce errors in the calculation of COD fractions, the value of fraction $X_S$ was calculated on the basis of the biochemical degradation rate determined in studies (k) for raw wastewater, whereas the $S_I$ fraction was calculated from the difference between SCOD and $BOD_{Tot}$ of filtered treated wastewater. $BOD_{Tot}$ of the treated wastewater was calculated taking into account the rate of biochemical degradation determined in the studies (k) for treated wastewater. The shares of individual COD fractions in raw wastewater calculated on the basis of the standard and modified procedure differed by approx. 10% in the case of suspension fractions. Modification of the methodology to determine the COD of the treated wastewater $S_S$ fraction significantly influenced the contents of all fractions in treated wastewater.

**Keywords:** COD fractions; biodegradability; wastewater characterization; municipal wastewater

---

## 1. Introduction

The basis for the design and optimization of biological treatment processes is a properly prepared balance of pollutants in raw wastewater. The effectiveness of these processes is determined by the content of biodegradable substances, which are a source of energy or a building material for microorganisms. The easily decomposable substances contained in the waste water include carbohydrates and proteins that are already metabolised as a result of catalytic and enzymatic reactions during transport through the wastewater system, which results in mainly the hydrolysis products of these compounds, i.e., amino acids and fatty acids, reaching the wastewater treatment plant. The distribution of slowly biodegradable waste, on the other hand, requires the presence of bacteria with a long generation time, the development of which is not possible using technologies used for easily decomposable substances [1–3]. The content of organic compounds in wastewater can be expressed indirectly as Total Organic Carbon (TOC) based on the amount of carbon dioxide produced during combustion [3,4]:

$$C_{org} + O_2 \xrightarrow{temp, catalyst} CO_2 \tag{1}$$

Another method is determining the content of organic compounds based on total oxygen demand (TOD). Determining TOD relies on the oxidation of organic substances at 950 °C in the presence of a catalyst. Under these conditions, decompose organic substances non-oxidized in the COD method.

Therefore, the value of TOD is higher than that of COD. Meanwhile, the ultraviolet absorbance ($UV_{250}$ nm) is a measure of the content only of organic compounds that absorb UV radiation, e.g., phenol, surface-active substances. A dependence was determined between COD and $UV_{250}$ represented by the formula Equation (2). [3]:

$$COD = 45.2 \cdot UV_{250} + 13.6 \tag{2}$$

The standard characteristic of wastewater in this area is usually based on the values of indicators such as Biochemical Oxygen Demand (BOD) and Chemical Oxygen Demand (COD).

Biochemical Oxygen Demand is a measure of the amount of biodegradable organic matter and determines the amount of oxygen needed for bacteria to oxidize biochemically degradable organic compounds under aerobic conditions, without access to light, at a temperature 20 °C [5,6]. The biochemical decomposition of organic compounds takes place in two phases: oxidation of carbon compounds and oxidation of nitrogen compounds. The kinetics of the first phase of BOD distribution can be described by the 1st order reaction:

$$BOD_t = BOD_{Tot} \cdot (1 - e^{-k \cdot t}) \tag{3}$$

where:

$BOD_t$—biochemical oxygen demand after time t,
$k$—coefficient of the rate of kinetic reaction,
$BOD_{Tot}$—total oxygen demand for the 1st phase of decomposition.

The course of oxygen consumption depends on many factors, among others on the type and concentration of organic substances, the amount and activity of microorganisms, the content of biogenic compounds, temperature, pH, and alkalinity, which affects the value of the $k$ coefficient, which in raw wastewater can range from 0.07 to 0.8 $d^{-1}$ [7].

In turn, the COD determination means an equivalent amount of oxygen taken from the oxidant (in mg $O_2$ $dm^{-3}$) needed to oxidize organic compounds and certain inorganic compounds to simple mineral forms [6,8]. The analytical method for determining COD can be schematically represented as:

$$C_xH_yO_z + \text{oxidizer} \xrightarrow{H_2SO_4 + catalyst + temperature} CO_2 + H_2O + \text{organiccompounds} \tag{4}$$

The efficiency of this reaction depends on: type of organic compounds, oxidizer and catalyst used, temperature, and heating time. The highest oxidation state of organic compounds is obtained using potassium dichromate in a strongly acidic environment and in the presence of iron salt as a catalyst. The presence of a catalyst is necessary for high concentrations of ammonia, organic amines or nitrogenous organic compounds. The catalyst for the oxidation reaction of alcohols and acids is $AgSO_4$. The COD index gives a good representation of the total organic content in wastewater. However, it should be remembered that numerous organic compounds are not determined under the conditions of this test, e.g., some aromatic (benzene, pyridine) and aliphatic (n-hexane, n-heptane) hydrocarbons. During the determination, it is possible to lose volatile organic compounds before they are oxidized. COD is a summary measure of the content of organic compounds in wastewater both subject to and not subject to biochemical degradation, but does not allow separate determination [9–11].

Although the processes of biochemical and chemical oxidation have a separate character, in a number of cases, especially for a given type of wastewater, there is a specific correlation between COD and $BOD_5$, which allows to initially estimate the biodegradability of pollutants. Wastewater can be considered susceptible to biodegradation if $1.5 < COD/BOD_5 < 2.5$. A high value of the $COD/BOD_5$ ratio (>2.5) indicates a slow decomposition and a high content of organic substances that are hardly decomposable or biologically indecomposable, which may be caused by a large proportion of industrial wastewater in municipal wastewater. In turn, the value of $COD/BOD_5 < 2.0$ indicates a significant content of biologically degradable contaminants [5,12,13] (Table 1).

**Table 1.** Biodegradability of organic compounds [12,13].

| COD/ BOD$_5$ | Value decrease COD, [%] | Assessment of Susceptibility to Biochemical Biodegradation |
|:---:|:---:|:---:|
| <2.0 | >90 | easily biodegradable |
| 2.0–2.5 | 50–90 | Biodegradable |
| 2.5–5.0 | 10–50 | slowly biodegradable |
| >5.0 | <10 | resistant to biodegradation |

Significant dependencies can be determined for wastewater which contains only substances undergoing biochemical oxidation and that are quantitatively oxidizable by chemical means. Apart from biologically easily biodegradable compounds, non-degradable, slowly degradable, or biodegradable compounds that are not oxidizable under the conditions of the dichromate method (e.g., pyridine and its derivatives) are present, this interpretation of results only has an illustrative dimension, and the practical evaluation of the composition wastewater is often misleading.

Knowledge of the share of organic contaminants susceptible to biodegradation and resistant to biodegradation is very important in the design of biological removal of nitrogen and phosphorus, as it influences the dynamics of the activated sludge process, e.g., the oxygen demand, maintaining a constant sludge age, and kinetic parameters of the biological reactor operation. Acceptance of COD as the main parameter determining the amount of organic carbon in wastewater and the division of COD into fractions describing different degrees of their biodegradation is for now a significant extension of the characteristic of wastewater [8]. One of the COD fractions plays an active component in biochemical transformations. Other factions are e.g., substrates in metabolism for the biomass. Among them, we distinguish easily biodegradable substrates and those that hydrolyse slowly. The rest of the organic matter is not biodegradable (inert). The division of COD in municipal wastewater into fractions is shown in Figure 1 [14].

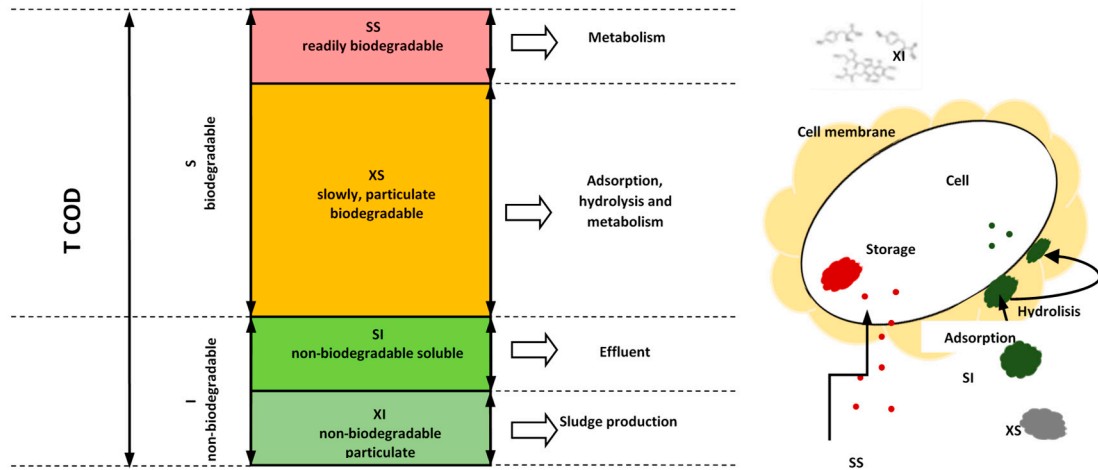

**Figure 1.** Schematic diagram of the chemical oxygen demand (COD) fractions and their fates in a biological wastewater treatment plant [14].

The total COD of wastewater, divided into fractions, can be calculated in a simplified way [15]:

$$TCOD = S_S + S_I + X_S + X_I, g\ O_2\ m^{-3} \tag{5}$$

where:

$S_S$—COD of soluble readily biodegradable substrates,

$S_I$—COD of inert soluble organic substrates,

$X_S$—COD of particulate slowly biodegradable substrates,

$X_I$—COD of inert particulate organic substrates,

and in detail [10]:

$$TCOD= S_{LKT}\ S_F + S_I + X_S + C_X + X_H + X_I, g\ O_2\ m^{-3} \tag{6}$$

where:

$S_{LKT}$—COD of volatile fatty acids,
$S_F$—COD of fermentable organic substrates,
$C_X$—COD of slowly biodegradable colloidal substrates,
$X_H$—COD of heterotrophic biomass fraction.

The COD fractionation method is not devoid of errors. The share of fractions can be determined in various methods and with varying accuracy. Some of the fractions cannot be determined directly [16]. $S_S$ fraction can be determined by filtration through a 0.1 or 0.45 µm filter, combining coagulation with $Zn(OH)_2$ and filtration through a 0.45 µm filter, based on tests of the oxygen uptake rate of the activated sludge, or most often with the use of $BOD_5$ measurements [1,17,18]. In the case of fraction $S_I$, only physico-chemical methods are known [16]. The fraction of inert particulate organic substrates ($X_I$) can be determined on the basis of the mass balance of the sludge in the system, taking into account active sludge, degradation products, and accumulation of fraction $X_I$ in the sludge accumulated in the secondary wastewater settling tank and lost with effluent [17] or by the direct weight method [19]. Concentration of fraction $X_S$ can be determined on the basis of the amount of oxygen used for the distribution of activated sludge, determined on the basis of batch or continuous tests.

The methods of determination of biodegradable fractions [1,17–19] described in the literature require a long time to obtain a result and additional analytical procedures deviating from standard determinations carried out in raw wastewater.

In the commonly used methodology for determining fractions according to ATV-A 131 [20], the fraction of particulate slowly biodegradable substrates $X_S$ is defined as the difference of total BOD, calculated on the basis of $BOD_5$ raw unfiltered wastewater and the rate of biochemical degradation and the immediately decomposed dissolved fraction on the basis of the assumed kinetic value of the decomposition rate $k$. The share of industrial or transported wastewater in a municipal wastewater system affects the rate of biochemical degradation of organic pollutants [21]. Therefore, the calculation of the COD fraction based on the assumed, and not determined for a given wastewater value of the coefficient of biochemical degradation rate $k = 0.1\ d^{-1}$ may be miscalculated.

Table 2 presents literary data on the share of individual COD fractions in municipal wastewater depending on the share of industrial wastewater.

The data comparison in Table 2 shows that the share of COD fraction in municipal wastewater is not constant and changes, sometimes considerably, and the proportion between individual factions is significantly affected by the share of industrial wastewater in them.

As describes above, fractionation and balancing of COD can be determined in various ways and with varying accuracy. In practice, good wastewater characteristics are obtained on the basis of COD fractionation in accordance with ATV-A 131 guidelines, especially when the wastewater characteristics are in high compliance with the assumptions of the method. The article proposes a modification of the ATV-A131 method that increases the accuracy of determining the COD fraction.

The conducted research tries to delve into the understanding of the different fractions of organic matter in wastewater, regarding its biodegradability. To use practical tests with a wider range of organic matter composition is required to be able to adopt the modified methodology. The results can be ground for further scientific discussion on this topic.

**Table 2.** Share of COD fraction in municipal raw wastewater and municipal wastewater with a significant share of industrial wastewater.

| | COD fractions, % | | | |
|---|---|---|---|---|
| $S_S$ | $S_I$ | $X_S$ | $X_I$ | |
| **Municipal wastewater** | | | | |
| 10–20 | 7–11 | 53–60 | 7–15 | Kappeler, Gujer, 1992 [19] |
| 9.0 | 4.0 | 77.0 | 10.0 | Sozen, 1998 [22] |
| 50.0–61.7 | 2.2–6.0 | 22.0–34.4 | 8.0–16.2 | Płuciennik-Koropczuk, Myszograj, 2017 [23] |
| 20–25 | 8–10 | 60–65 | 5–7 | Ekama, 1986 [15] |
| 24–32 | 8–11 | 43–49 | 11–20 | Henze, 2002 [17] |
| **Municipal wastewater with a significant share of industrial wastewater** | | | | |
| | | Textile industry wastewater | | |
| 25.0 | 14.0 | 59.0 | 2.0 | Baban et al. 2004 [24] |
| | | Dairy industry wastewater (10%) | | |
| 38.8 | 2.3 | 45.5 | 14.8 | Struk-Sokołowska, 2015 [25] |
| | | Baking industry wastewater (10%) | | |
| 38.8 | 1.0 | 44.2 | 15.2 | Struk-Sokołowska, 2017 [26] |
| | | Oil processing wastewater | | |
| 29.2 | 9.9 | 37.4 | 23.5 | Chiavola 2014 [27] |
| | | Paper industry wastewater (25%) | | |
| 4.2 | 39.5 | 43.1 | 13.2 | Choi 2017 [14] |

## 2. Aims and Methodology of the Study

The aim of the research was to determine the share in municipal wastewater, raw and treated, of waste resistant and susceptible to biodegradation on the basis of COD fraction determined in accordance with standard and modified assumptions.

The methodology for determining the COD fraction has been included in the ATV-A131 guidelines [20]. Guideline ATV-DVWK A131P *The dimensioning of single-stage sewage treatment plants with activated sludge,* developed by the German Association of Engineers and Technicians of Wastewater Treatment, is widely used in design practice and at the stage of optimization and modeling of the wastewater treatment process. It allows to describe processes based on wider than basic characteristics of raw wastewater. According to which, the determination of the $S_S$, $S_I$, $X_S$ and $X_I$ fraction is based on the determination of COD and $BOD_5$ in samples of filtered (0.45 μm) and unfiltered raw and treated wastewater, where:

- The COD of inert soluble organic substrates $S_I$ is defined as COD in treated filtered wastewater.
- The COD of soluble readily biodegradable substrates $S_S$ is calculated from the difference in the concentration of dissolved organic pollutants $S_{COD}$ determined in raw filtered wastewater and fraction $S_I$: $S_S = S_{COD} - S_I$.
- The COD of particulate slowly biodegradable substrates $X_S$ is defined as the difference of total BOD, calculated on the basis of $BOD_5$ raw unfiltered wastewater and the rate of biochemical degradation and the easily decomposed dissolved fraction: $X_S = (BOD_5 / k1) - S_S$.
- The fraction of inert particulate organic substrates XI is determined from the dependence: $X_I = X_{COD} - X_S$ where $X_{COD}$ is the total COD of organic suspensions.
- The total COD of wastewater is the sum of all fractions: $TCOD = S_I + S_S + X_S + X_I$.

According to the methodology ATV-131, the value of fraction $X_S$ is calculated on the basis of the constant rate of biochemical degradation k = 0.1 $d^{-1}$, for which $BOD_{Tot} = BOD_5 / 0.6$. The calculation also assumes that the fraction $S_I$ corresponds to the COD value in filtered treated wastewater. This assumption is reflected in the calculation of the $S_S$ fraction because it is calculated based on the difference $S_{COD} - S_I$, which in treated wastewater is equal to zero in each case.

In order to reduce errors in the calculation of COD fractions resulting from the simplifications used in the ATV-131 methodology, the value of fraction $X_S$ in the modified method was calculated on the basis of the biochemical degradation rate determined in studies (*k*) for raw wastewater, whereas the $S_I$ fraction was calculated from the difference between $S_{COD}$ and $BOD_{Tot}$ of filtered treated wastewater. $BOD_{Tot}$ of the treated wastewater was calculated taking into account the rate of biochemical degradation determined in the studies (*k*) for treated wastewater.

To determine the dissolved fraction, the method of filtering the wastewater samples through the 0.45 μm filter was assumed.

Samples of wastewater for testing were collected from a mechanical–biological wastewater treatment plant with an average daily flow of 3450 m$^3$ d$^{-1}$ (approx. 27,000 P.E.), working in the technology of low-load activated sludge. Wastewater is supplied to the wastewater treatment plant by sanitary sewerage and transported by septic tankers.

## 3. Results

The characteristics of organic pollutants in raw and treated wastewater are presented in Table 3. During the tests carried out on the raw and treated wastewater samples, high variability of indicators expressing the content of organic pollutants was not noted. In raw wastewater, the average values were: COD = 1164 ± 157 mg O$_2$ dm$^{-3}$, BOD$_5$ = 641 ± 84 mg O$_2$ dm$^{-3}$, and TOC = 267 ± 93 mg O$_2$ dm$^{-3}$. The average efficiency of removal of organic substances was 94.6 ± 1.5% for COD, 97.8 ± 0.7% for BOD$_5$, and 94.5 ± 1.8% for TOC.

**Table 3.** Characterization of organic waste in raw and treated wastewater.

| Wastewater Sample | Range | COD mg O$_2$ dm$^{-3}$ | BOD$_5$ mg O$_2$ dm$^{-3}$ | TOC mg C dm$^{-3}$ |
|---|---|---|---|---|
| raw | minimum | 1000 | 548 | 182.7 |
| | maximum | 1340 | 738 | 451.3 |
| | average | 1164 ± 157 | 641 ± 84 | 267 ± 93 |
| treated | minimum | 40 | 7.7 | 5.7 |
| | maximum | 98 | 24 | 22.1. |
| | average | 62 ± 22 | 14 ± 6 | 14 ± 5 |

Values of the COD/BOD$_5$ quotient for raw wastewater were from 1.8 to 2.0, which at the efficiency of COD removal above 90% classifies organic contaminants contained in wastewater entering the treatment plant as easily biodegradable (Table 1).

The value of the quotient (COD/BOD$_5$ = 3.6–6.2) calculated for treated wastewater indicates that the compounds were slowly biodegradable (COD/BOD$_5$ = 3.6–5) or non-biodegradable (COD/BOD$_5$ > 5–6.2). Estimated on the basis of changes in the COD/BOD$_5$ value, susceptibility to biodegradability of wastewater has been extended by the COD fraction values expressing susceptible and biodegradable fractions, taking into account the presence of contaminants (dissolved, undissolved).

The separation of the fractions into dissolved and undissolved was performed by filtration through a 0.45 μm. filter. In order to control the degree of fraction separation, the particle sizes were determined before and after the filtration process. The distribution of particle sizes in samples of non-filtered wastewater and after filtration is shown in Figure 2.

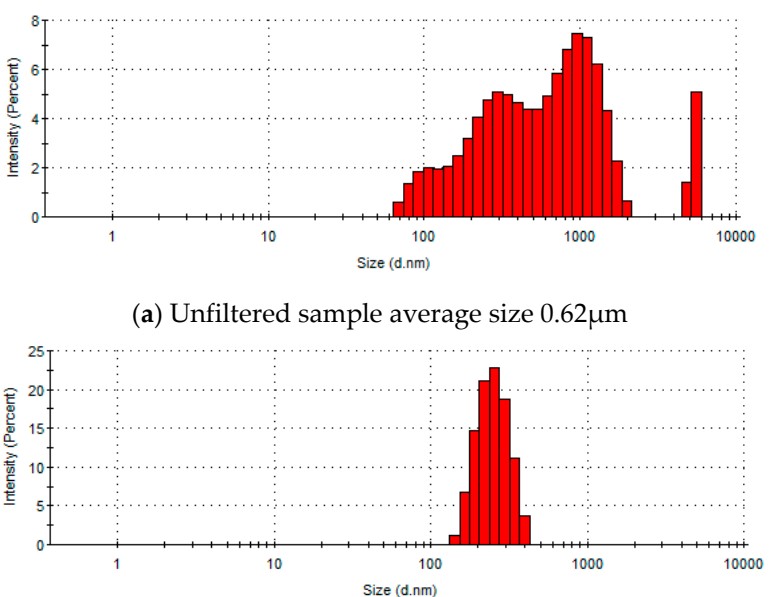

(**a**) Unfiltered sample average size 0.62µm

(**b**) Filtered sample average size 0.29 µm

**Figure 2.** Particle size distribution in raw wastewater samples (**a**) and after filtration through 0.45 µm filter (**b**).

The particles of sizes between 0.22 and 1.72 µm were found most frequently in the raw wastewater sludge, and of sizes between 0.16 to 0.30 µm in the filtered wastewater. The results confirm the appropriateness of the fraction separation by filtration on a 0.45 µm membrane.

Table 4 presents the values of individual fractions and their percentage in total COD of raw and treated wastewater, determined in accordance with the standard ATV-A131 methodology.

**Table 4.** COD fractions in raw wastewater and treated according to the ATV-A131 method.

| Wastewater Sample | Value | $S_S$ | $S_I$ | $X_S$ | $X_I$ | Total COD |
|---|---|---|---|---|---|---|
| | | | | mg $O_2$ dm$^{-3}$ | | |
| | Minimum | 230 | 20 | 663 | 47 | 1000 |
| Raw | Maximum | 352 | 48 | 990 | 95 | 1340 |
| | Average | 280.4 ± 56.3 | 34.4 ± 10.4 | 787.6 ± 131.8 | 61.2 ± 19.2 | 1164 ± 157 |
| | Minimum | 0 | 20 | 13 | 2 | 40 |
| Treated | Maximum | 0 | 48 | 40 | 10 | 98 |
| | Average | 0 ± 0.0 | 34.4 ± 10.4 | 23.4 ± 10.2 | 4.6 ± 3.2 | 62 ± 22 |
| | | % | | | | |
| | Minimum | 18.2 | 1.9 | 62.8 | 4.2 | |
| Raw | Maximum | 29.5 | 4.0 | 75.1 | 7.1 | |
| | Average | 24.2 ± 4.3 | 3.0 ± 0.9 | 67.6 ± 5.3 | 5.2 ± 1.1 | (-) |
| | Minimum | 0 | 49.0 | 27.1 | 5.0 | |
| Treated | Maximum | 0 | 66.7 | 45.0 | 10.2 | |
| | Average | 0 ± 0 | 55.9 ± 7.2 | 37.3 ± 7.0 | 6.8 ± 2.1 | |

In raw wastewater, the highest COD values were noted for biodegradable fractions: COD of slow-biodegradable undissolved fraction ($X_S$) was on average 787.6 ± 131.8 mg $O_2$ dm$^{-3}$, and COD of the dissolved fraction $S_S$ 280.4 ± 56.3 mg $O_2$ dm$^{-3}$. The average COD values of non-biodegradable, suspended and dissolved fractions were $X_I$ = 61.2 ± 19.2 mg $O_2$ dm$^{-3}$, and $S_I$ = 34.4 ± 10.4 mg $O_2$ dm$^{-3}$, respectively. The share of individual fractions in the total COD was as follows: $X_S$ = 67.6 ± 5.3%, $S_S$ = 24.2 ± 4.3%, $X_I$ = 5.2 ± 1.1% and $S_I$ = 3.0 ± 0.9%.

In treated wastewater, the average COD values of the determined fractions were: $S_I$ = 34.4 ± 10.4 mg $O_2$ dm$^{-3}$, $X_S$ = 23.4 ± 10.2 mg $O_2$ dm$^{-3}$ and $X_I$ = 4.6 ± 3.2 mg $O_2$ dm$^{-3}$, and the corresponding share in the total COD was as follows: $S_I$ = 55.9 ± 7.2%, $X_S$ = 37.3 ± 7.0% and $X_I$ = 6.8 ± 2.1%. According to

the standard methodology for determining the COD fraction in treated wastewater, no SS fraction was recorded.

In order to calculate specific fractions of COD based on the modified methodology, the rates of raw and treated wastewater were determined for the rate of biochemical degradation ($k$). The course of the BOD curves on the basis of which the constants ($k$) are determined for raw and treated wastewater are shown in Figure 3.

$$BOD_t = BOD_{Tot} (1 - e^{-k \cdot t})$$

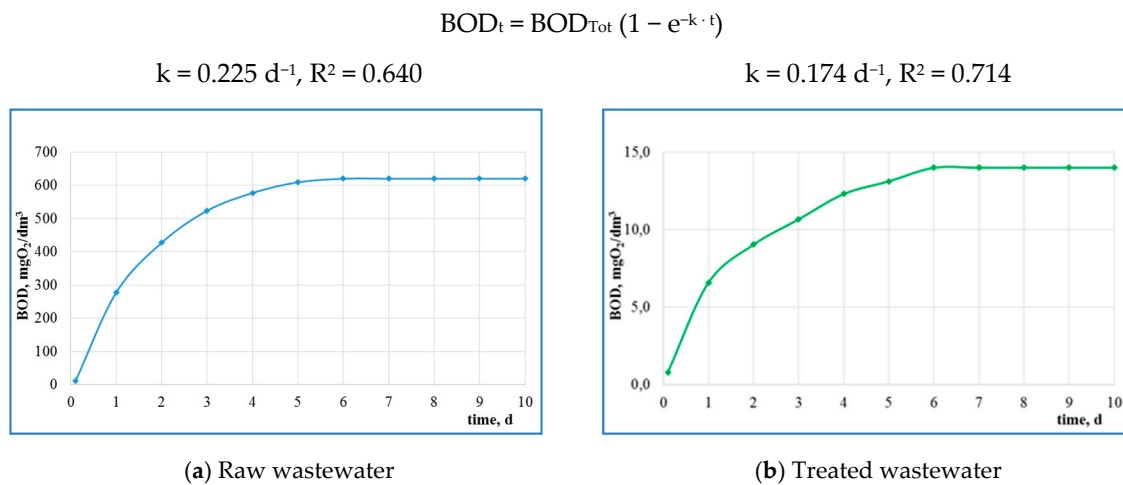

$k = 0.225\ d^{-1}, R^2 = 0.640$　　　　　　　　　　　　　　　$k = 0.174\ d^{-1}, R^2 = 0.714$

(**a**) Raw wastewater　　　　　　　　　　　　　　　(**b**) Treated wastewater

**Figure 3.** The course of BOD curves for raw and treated wastewater. (**a**) Raw wastewater; (**b**) treated wastewater.

The biochemical degradation rate values determined for raw wastewater and treated wastewater differ significantly from the value of constant $k = 0.1\ d^{-1}$, taken as a standard in calculations in accordance with ATV-A131. The mean values of the biochemical degradation rate constants were, respectively, for raw wastewater $k = 0.225\ d^{-1}$ and for treated wastewater $k = 0.174\ d^{-1}$, which corresponds to raw wastewater $BOD_{Tot} = BOD_5 / 0.68$, and in treated $BOD_{Tot} = BOD_5 / 0.58$.

The COD values of individual fractions determined on the basis of the modified ATV-A131 methodology and the percentage of fractions in total COD of raw and treated wastewater are presented in Table 5.

**Table 5.** COD fractions in raw wastewater and treated wastewater according to the modified ATV-A131 method.

| Wastewater sample | Value | $S_S$ | $S_I$ | $X_S$ | $X_I$ | Total COD |
|---|---|---|---|---|---|---|
| | | | | mg O$_2$ dm$^{-3}$ | | |
| Raw | Minimum | 243 | 7 | 545 | 165 | 1000 |
| | Maximum | 359 | 41 | 845 | 236 | 1340 |
| | Average | 292 ± 55.4 | 22.8 ± 12.5 | 656.3 ± 121.5 | 192.5 ± 27.8 | 1164 ± 157 |
| Treated | Minimum | 7 | 7 | 3 | 12 | 40 |
| | Maximum | 16 | 41 | 29 | 24 | 98 |
| | Average | 11.7 ± 4.4 | 22.8 ± 12.5 | 10.9 ± 9.3 | 18.7 ± 4.6 | 62 ± 22 |
| | | % | | | | |
| Raw | Minimum | 18.7 | 0.7 | 51.2 | 15.2 | |
| | Maximum | 30.8 | 3.1 | 64.1 | 17.6 | |
| | Average | 25.3 ± 4.5 | 1.9 ± 0.9 | 56.3 ± 5.3 | 16.5 ± 0.9 | (-) |
| Treated | Minimum | 16.7 | 6.9 | 6.5 | 21.7 | |
| | Maximum | 51.2 | 33.3 | 29.3 | 42.6 | |
| | Average | 23.4 ± 10.2 | 35.0 ± 13.2 | 12.5 ± 9.5 | 31.6 ± 8.5 | |

In raw wastewater, the highest COD values were recorded for biodegradable fractions and were respectively $X_S = 656.3 ± 121.5$ mg O$_2$ dm$^{-3}$ and $S_S = 292 ± 55.4$ mg O$_2$ dm$^{-3}$, while the average COD of non-biodegradable, suspended and dissolved fractions were at the level of: $X_I = 192.5 ± 27.8$

mgO$_2$/dm$^3$ and S$_I$ = 22.8 ± 12.5 mg O$_2$ dm$^{-3}$. The share of individual fractions in the total COD was as follows: X$_S$ = 67.6 ± 5.3%, S$_S$ = 24.2 ± 4.3%, X$_I$ = 5.2 ± 1.1% and S$_I$ = 3.0 ± 0.9%. In treated wastewater, the average COD values for the analyzed fractions were: S$_I$ = 22.8 ± 12.5 mg O$_2$ dm$^{-3}$ and X$_S$ = 9.3 ± 10.9 mg O$_2$ dm$^{-3}$, S$_S$ = 11.7 ± 4.4 mg O$_2$ dm$^{-3}$, X$_I$ = 18.7 ± 4.6 mg O$_2$ dm$^{-3}$, and the corresponding share in the total COD was as follows: S$_I$ = 35 ± 13.2%, S$_S$ = 20.9 ± 10.2%, X$_S$ = 12.5 ± 9.5% and X$_I$ = 31.6 ± 8.5%.

## 4. Discussion of Results

Based on the conducted research, it was shown that the method of assessing wastewater susceptibility to biochemical degradation and assumptions accepted for calculations have a significant impact on the obtained results. Figures 4 and 5 present the total and detailed share of biodegradable and non-biodegradable COD fractions in raw and treated wastewater, determined on the basis of the standard assumptions of ATV-A131 and the modified calculation methodology. In the raw wastewater, biodegradable fractions (COD-S) accounted for 91.8 and 81.6% of total COD, respectively, and non-biodegradable (COD-I) 8.2 and 18.4% (difference by approx. 10%). In treated wastewater, the difference in fraction content depending on the calculation methodology was approx. 3%. The biodegradable fractions equaled 37.3 and 33.4% respectively, with the share of non-biodegradable fraction being 62.7 and 65.6%.

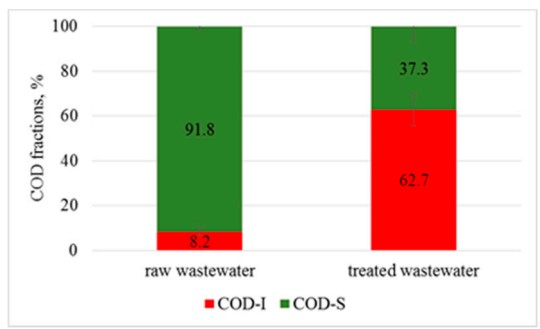 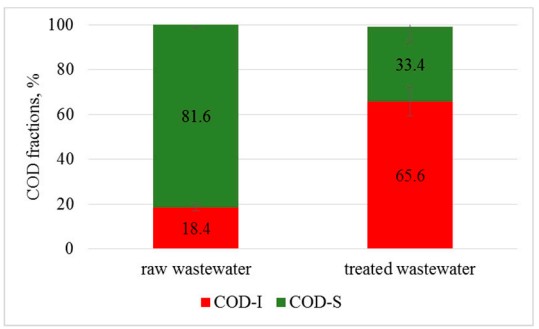

(**a**) Methodology in accordance with ATV-A131　　　　(**b**) Modified calculation methodology

**Figure 4.** Total share of COD biodegradable and non-biodegradable fraction in raw and treated wastewater. (**a**) Methodology in accordance with ATV-A131; (**b**) modified calculation methodology.

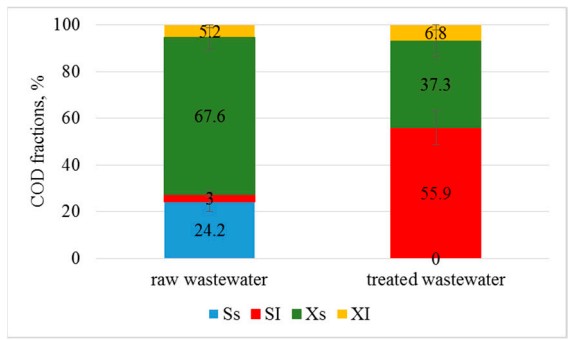 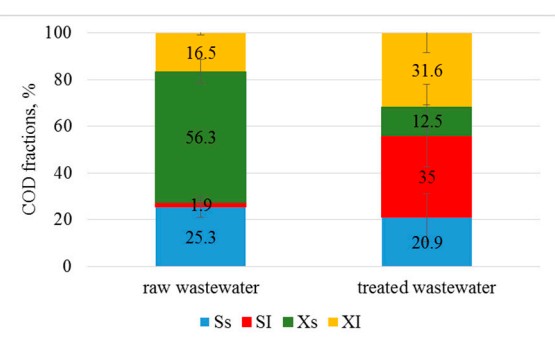

(**a**) Methodology in accordance with ATV-A131　　　　(**b**) Modified calculation methodology

**Figure 5.** Average fraction shares in total COD of raw and treated wastewater. (**a**) Methodology in accordance with ATV-A131; (**b**) modified calculation methodology.

A detailed analysis of the basic values of the COD fraction showed that in the raw wastewater, the highest COD was obtained for the fractions: X$_S$ and S$_S$ (biodegradable fractions), and X$_I$ and S$_I$ (non-decomposed fractions) (Figure 5). The shares of the dissolved fractions were slightly different, and equalled respectively S$_S$ and S$_I$: 24.2 and 3%, and 25.3 and 1.9%. However, the difference for

the COD value of the suspension fraction was over 10%. The shares of this fraction are respectively $X_S = 67.6\%$ and $X_I = 5.2\%$, and $X_S = 56.3\%$ and $X_I = 16.5\%$.

According to the standard method of determining the ATV-A13 fraction in treated wastewater, the proportion of individual fractions was as follows: $S_I = 55.9\%$, $X_S = 37.3\%$ and $X_I = 6.8\%$. Modification of the methodology made it possible to determine the share in COD of wastewater of the treated $S_S$ fraction, which significantly affected the share of all factions that were at the level of: $S_I = 35\%$, $S_S = 20.9\%$, $X_S = 12.5\%$ and $X_I = 31.6\%$.

## 5. Conclusions

Depending on the adopted method, the susceptibility of contaminants contained in wastewater to biodegradation can be estimated with varying accuracy. Initial estimation of wastewater biodegradability based on the value of the COD/BOD$_5$ ratio in raw and treated wastewater, combined with a COD removal, allows to control the correctness of the biochemical process of oxidation of organic pollutants contained in wastewater. However, it does not provide information on the share of susceptible and biodegradable compounds in wastewater. The division of COD into fractions is a significant extension of wastewater characteristic in terms of its degree of biodegradation. The methods of determination of biodegradable fractions described in the literature require a long time to obtain a result and additional analytical procedures deviating from standard determinations carried out in raw wastewater.

Good wastewater characteristics are obtained based on COD fractionation in accordance with ATV-A 131 guidelines. The method of determining individual fractions is based on the adoption of general calculation assumptions, which, for example, does not allow to determine in the treated wastewater the proportion of biodegradable dissolved pollutants (COD of fraction $S_S$).

Introducing to the ATV-A131 calculation methodology a modification regarding the determination of concentrations of fractions $X_S$ and $S_I$ while taking into account the kinetic coefficients determined for raw and treated wastewater makes it possible to determine the shares of individual fractions with greater accuracy. Calculation of the $S_I$ fraction concentration from the difference between $S_{COD}$ and $BOD_{Tot}$ of the treated filtered wastewater on the basis of a constant decomposition rate allows to determine the concentration of the $S_S$ fraction in treated wastewater, which in turn affects the value of share of other fractions.

The advantage of the modified method is the possibility of using it to characterize raw wastewater in the design and optimization of domestic wastewater treatment plants, based on one-off tests. The disadvantage is the inability to use in the case of wastewater, which characteristics differ significantly from the average values assumed in ATV-A131, e.g., for wastewater with a large share of industrial wastewater, inference about their biodegradability based on COD fraction calculated in accordance with this methodology may be incorrect.

**Author Contributions:** Conceptualization, E.P.-K. and S.M.; Methodology, E.P.-K. and S.M.; Software, E.P.-K. and S.M.; Validation, E.P.-K. and S.M.; Formal Analysis, E.P.-K. and S.M.; Investigation, E.P.-K. and S.M.; Resources, E.P.-K. and S.M.; Writing-Original Draft Preparation, E.P.-K. and S.M.; Writing-Review & Editing, E.P.-K. and S.M.; Visualization, E.P.-K. and S.M.; Supervision E.P.-K. and S.M.

**Funding:** This research received no external funding.

**Conflicts of Interest:** The authors declare no conflict of interest.

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
