# Peer review of "New Approach in COD Fractionation Methods"

_water, doi:10.3390/w11071484_

Round 1

Reviewer 1 Report

This study tries to deepen into the understanding of the different fractions of organic matter in wastewater, regarding its biodegradability. The methodology and the approach that the authors followed seems reasonable. I believe the results should be published and made available to the wider scientific audience, to enable further discussion on the topic. It is clear that more practical tests with a wider range of OM composition is required to be able to adopt the modified  methodology.

Author Response

Dear Reviewer.

Thank you for your comments. All of them were taken into account and an amendment was made to the manuscript. Changes are made using the "track changes".

Reviewer 2 Report

Topic of the manuscript is very relevant to the journal and for the real world as well. This will give a new powerful insight to assess WWTP's efficiency.

Authors did a fantastic job to present their research findings in articulate manner. Before being accepted for publication, i have couple of comments:

Considering significant amount of grammatical errors, English proofreading for this manuscript is a must before being accepted for publication.   

Please explain ATV-A 131 guideline in the methodology part rather just giving a reference for the same to make it clear how your modification going to help.

Author Response

(The authors gave the same response as above.)
